# Techno-Economic and Environmental Analysis of a Hybrid PV-WT-PSH/BB Standalone System Supplying Various Loads

**Mohammed Guezgouz [1,\*], Jakub Jurasz [2,3,\*]** 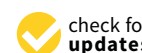 **and Benaissa Bekkouche [1]**

[1] Department of Electrical Engineering, Mostaganem University, BP188/227, Mostaganem 27000, Algeria; bekbenm@yahoo.fr
[2] School of Business, Society and Engineering, Future Energy Center, Mälardalen University, 72123 Västerås, Sweden
[3] Department of Engineering Management, Faculty of Management, AGH University of Science and Technology, 30059 Cracow, Poland
[\*] Correspondence: mohammed.guezgouz@univ-mosta.dz (M.G.); jakubkamiljurasz@gmail.com (J.J.)

**Abstract:** The Algerian power system is currently dominated by conventional (gas- and oil-fueled) power stations. A small portion of the electrical demand is covered by renewable energy sources. This work is intended to analyze two configurations of renewables-based hybrid (solar–wind) power stations. One configuration was equipped with batteries and the second with pumped-storage hydroelectricity as two means of overcoming: the stochastic nature of the two renewable generators and resulting mismatch between demand and supply. To perform this analysis, real hourly load data for eight different electricity consumers were obtained for the area of Mostaganem. The configuration of hybrid power stations was determined for a bi-objective optimization problem (minimization of electricity cost and maximization of reliability) based on a multi-objective grey-wolf optimizer. The results of this analysis indicate that, in the case of Algeria, renewables-based power generation is still more expensive than electricity produced from the national grid. However, using renewables reduces the overall $CO_2$ emissions up to 9.3 times compared to the current emissions from the Algerian power system. Further analysis shows that the system performance may benefit from load aggregation.

**Keywords:** hybrid system; renewable energy; solar–wind; pumped storage hydro-electricity; battery; Multi-Objective grey wolf optimizer (MOGWO); techno-economic analysis; $CO_2$ emission

## 1. Introduction

Numerous countries have changed their policies and adopted alternative energy sources because of the depletion of fossil fuels and the worrying wave of global warming, alongside the increasing costs of energy [1,2]. To meet these multiple challenges, Algeria has announced, through its state-managed energy company Sonelgaz, the implementation of three projects, namely the national renewable energy development program (2015–2030); the energy efficiency program and another that concerns twenty villages in southern Algeria [1,3]. The objective would be to achieve 22 GW of installed capacity by 2030; of which solar and wind energy would account for up to 13,575 MW and 5010 MW, respectively [4]. Moreover, these programs with a combined share of 37% of the installed capacity will cover 27% of the national electricity generation demand [3]. Considering the random and stochastic nature of these renewable resources [2], the implementation of such projects will alone not be sufficient to reach these targets. One of the promising options to increase renewable penetration requires efficient use of energy storage technologies; this should ensure a smooth transition from an energy system dominated by conventional power stations to one based on renewables [5].

Due to the manufacturing effect of scale and technological progress, the price gap between renewables-based electricity and that from conventional generators has decreased significantly [6,7] or has even been inverted in relation to older conventional generators. Considering this and the growing environmental awareness, research into low-carbon power technologies has gained much importance in recent years [6,8]. However, there is a need to be more accurate in the size and evaluation of such systems [6,8]. To identify the current state of knowledge about hybrid renewable energy systems, a relevant literature review along with a description of the analytical scope of the paper will be presented in the next subsection.

## 1.1. Related Works

In recent years, numerous research papers have been dedicated to the analysis of hybrid renewables-based energy sources [6,9,10]. This interest results from the fact that both climate and weather-driven energy sources tend to exhibit distinctive temporal and spatial patterns. Nevertheless, if exploited properly they have the potential to increase system reliability and/or decrease overall energy cost [11,12].

This paper is dedicated to hybrid systems utilizing solar (PV) and wind (WT) energies coupled with energy storage in the form of a battery bank (BB) or pumped-storage hydroelectricity (PSH). Consequently, two topologies are explored: PV–WT–BB and PV–WT–PSH. Both are assumed to be operating in an off-grid mode.

The concept of coupling wind and solar sources with PSH is especially popular in the context of island communities. Inhabited islands usually have significant potential in renewable resources but are also suited to the installation of PSH facilities. For PSH, an upper and lower reservoir are required to be located at different altitudes (and many islands have such sites). A study by Ma et al. [13] investigated the technical feasibility of a PV–WT–PSH system supplying an island community in Hong Kong. In their follow-up study, Ma et al. [14] proposed an optimal design for such a station. In both cases, the considered load was relatively low and averaged 250 kWh/day. In the case of another island community, Petrakopolou et al. [15], too, considered a PV–WT–PSH system with an interesting option to use any energy surpluses that appeared from PV and WT generation to produce hydrogen, which can later be used in various processes or converted back to electrical energy. The load considered by Petrakopolou et al. [15] amounted to 15.53 GWh/year, with a peak demand of 4.67 MW. An interesting study by Chen et al. [16] considered an island with over 100,000 inhabitants for which the observed load ranged from 15 MW to 50 MW, with a peak observed during the summer months [16], which aimed at reducing the diesel engine usage by optimizing the parameters of the PV–WT–PSH system. A study by Pérez-Díaz et al. [17] considered a case study of a large island community whose load in 2012 varied from 300 MW to 500 MW. The authors investigated the potential of a PSH station to reduce the system scheduling cost when large-scale wind sources are utilized. Notton et al. [18], on the other hand, investigated an island system characterized by a daily demand (during summer) of 5.6 GWh and, based on analysis of scenarios, showed how a set of reversible pumps operating within PSH can reduce the peak load. In the considered case, the load was covered both from conventional generators and from wind and solar sources. A research paper by Wang et al. [19] presented an interesting concept, according to which, small hydropower stations are used to smooth the power output of variable solar and wind generators, whereas PSH is used to shift load. A study by Garcia Latorre et al. [20] evaluated a well-known case study of the El Hierro island (with an annual consumption of 43.6 GWh), in which, adding the WT–PSH station to the previous system (conventional generators) increased the electricity cost, but simultaneously reduced emissions and improved supply reliability. A very recent paper by El Tawil et al. [21] considered a WT–PSH hybrid coupled with tidal energy and diesel generators. The system-covered load ranged from 0.4 MW to 1.4 MW and exhibited a peak during wintertime. The optimization proposed by the authors focused on reducing the cost of electricity and $CO_2$ emissions.

Meanwhile, the combination of PV and WT with battery storage is a much older concept that can be traced back to the advent of commercially available solar and wind technologies. Such systems

are especially popular for the off-grid operation mode but, over recent years (mainly due to the rapid reduction of the battery costs), their cost-effectiveness in the on-grid mode has been improving.

Regarding PV–WT–BB systems, a study by Borhanazad et al. [22] scrutinized their techno-economic performance for three typical rural areas. The objective function aimed at maximizing reliability, reducing the cost of electricity and increasing the share of renewables. The considered loads were relatively small and ranged from 0.5 kW to 5 kW. A long-term analysis was performed by Ogunjuyigbe et al. [23], in which a PV–WT–BB system coupled to a diesel generator was used to supply the load of a small household. The objective, again, aimed at minimizing the cost of electricity, $CO_2$ emissions and curtailed renewable generation (i.e., energy that is generated by renewables but that is in excess of what the grid can accept, and thus goes to waste). Kaabeche et al. [7] proposed a method to optimize the size of a PV–WT–BB hybrid system based on a firefly algorithm with a view to minimizing the cost of electricity for a given dissatisfaction rate (reliability) of a residential household. Ghorbani et al. [24] analyzed the potential of PV–WT–BB systems to cover the residential load assuming the minimization of electricity cost and increased reliability. A specific load in the form of telecom towers was investigated by Junaid et al. [25]. The authors considered various hybrid systems, including PV–WT–BB, aiming at minimizing the cost of electricity for different cities in India. An important study from the perspective of this paper is one by Yahiaoui et al. [26], who considered a PV–WT–BB system for supplying the load of a rural village in Southern Algeria. The system has been optimized by means of the grey-wolf optimizer (GWO) showing its superiority over the particle swarm optimizer (PSO). An interesting study by Al-Gussain et al. [27] considered supplying part of a cement factory from a PV–WT–BB system in order to maximize the positive environmental effect (reducing $CO_2$ emissions) and minimize the electricity bill. Kamal et al. [28] inspected another interesting load in the form of a bitumen tank. The objective of the PV–WT–BB system was to maintain a required temperature in the tank, which required the provision of electrical power ranging from 1.2 kW to 1.6 kW. A good summary of the works of PV–WT–BB systems is provided by Al-Falahi et al. [6], who conclude that although the operation of those systems is most often supported by diesel generators, this is not the environmentally friendly option.

### 1.2. Paper Contribution

Prior research articles have had several common features. Most researchers used hourly time series of relevant parameters, usually covering one meteorological year. The optimization is performed based on meta-heuristic methods such as a genetic algorithm (GA), particle swarm optimization (PSO), grey-wolf optimizer (GWO), software tools or scenario analysis (most often when one of the system components already exists). The optimization problem is usually multi-objective, including criteria such as reliability, cost of energy, share of renewables or fuel consumption reduction. The most common load is residential (a group of houses or individual houses) or an aggregated load of a bigger set of energy consumers (especially in the case of island communities). However, several studies address specific loads such as cement factories, telecom towers or bitumen tanks.

Considering the aforementioned references, the main contribution of this paper is its comprehensive comparison of a hybrid PV–WT system with different storage technologies, namely pumped storage hydro-electricity and battery bank, to meet the demand of various types of loads in a single location. Recently, ref [29] proposed a new meta-heuristic algorithm, called a multi-objective grey-wolf optimizer (MOGWO). This was applied to solve the optimal size for the proposed systems with regard to its reliability and its cost of energy. The study shows how the characteristics of different loads affect the system performances and evaluates what the potential benefits of aggregating loads are. For all systems, a sensitivity analysis was carried out with regard to changing solar- and wind-resource availability based on a ten-year, historical meteorological data.

## 2. Hybrid System Modeling and Simulation

Figure 1 illustrates the conceptual designs of the considered hybrid PV–WT–PSH and PV–WT–BB systems. (The systems are not connected to the mains power network.) The site selected to conduct this study is located in the North West of Algeria, more specifically the region of Mostaganem. The geographical coordinates of this site are (35°59′40.5″ N 0°08′27.2″ E, 380 m). The site is characterized by a mountain of more than 380 m altitude and the availability of space to install large-scale renewable generators. In order to reduce the cost of construction, the sea is assumed as the lower reservoir for the PSH station.

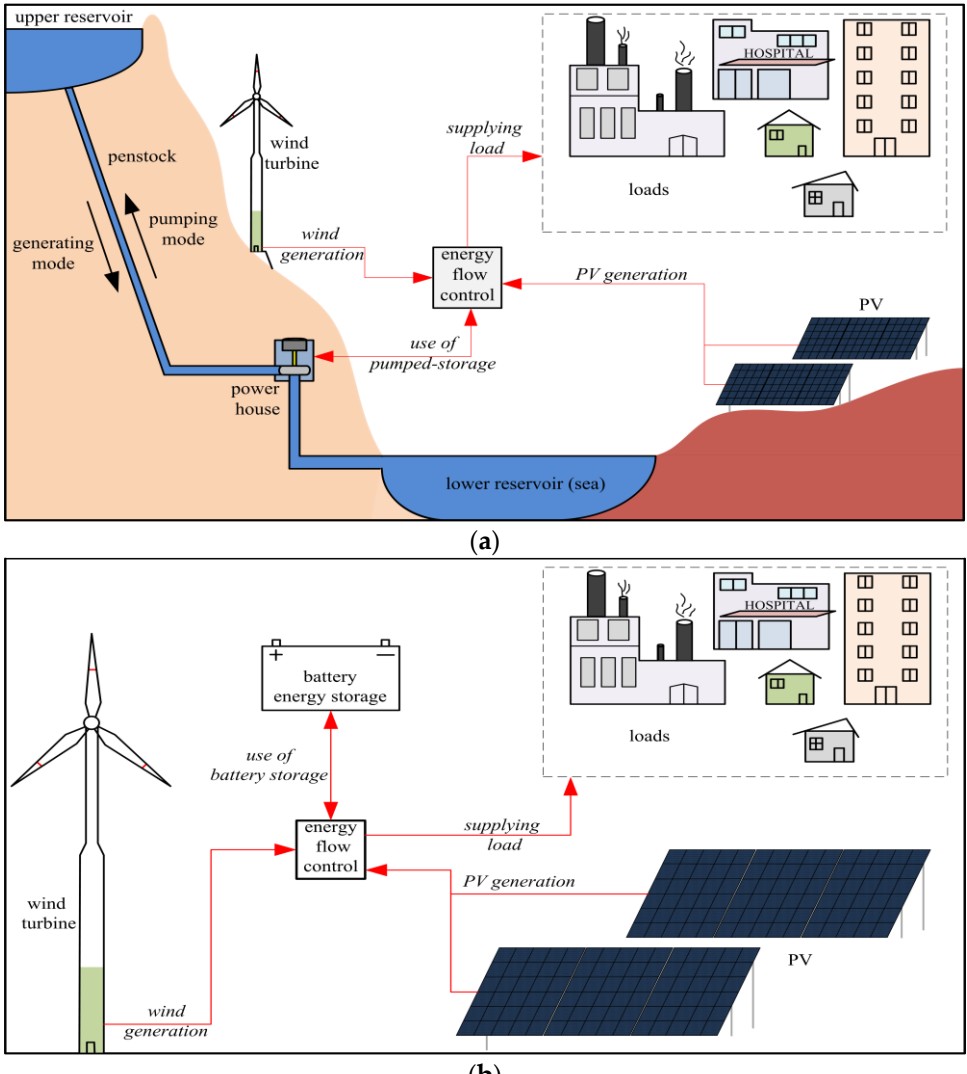

**Figure 1.** (**a**): Conceptual schematic design of the proposed standalone hybrid PV–WT–PSH system. (**b**): Conceptual schematic design of hybrid PV–WT–BB system in off-grid mode.

### 2.1. PV System Model

The instant output power delivered from a PV array is given by [22,30] in the Equation (1), where $P_{rate\_pv}$ is the rated power under standard test conditions, $\eta_{pv}$ is the overall loss factor, which represents all the losses due to the tilted angle, shading, inverter, wire losses and all other components [31]:

$$P_{pv}(t) = P_{rate\_pv} \times \frac{G}{G_{ref}} \times \left[1 + K_t\left(T_{cell} - T_{ref}\right)\right] \times \eta_{pv} \tag{1}$$

where $G$ is the hourly solar radiation and $G_{ref}$ is 1000 W/m². $T_{ref}$ is equal to 25 °C and $T_{cell}$ is the reference cell temperature in Equation (2), given by [31]:

$$T_{cell} = T_{amb} + \left(\frac{NOCT - 20}{800}\right) \times G \tag{2}$$

where NOCT is normal cell operating temperature (°C), $T_{amb}$ ambient temperature (°C).

The energy yield from a PV generator system essentially depends on the intensity of solar irradiation and temperature of the selected site [32]. The data regarding irradiation and temperature considered in this study, as shown in Figure 2, was collected from Copernicus Atmosphere Monitoring Service (CAMS) [33].

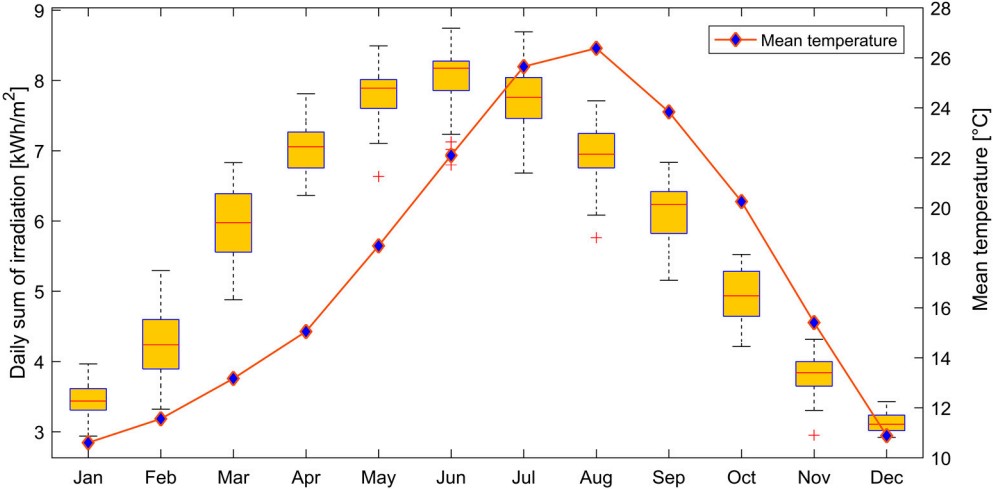

**Figure 2.** Daily sum of global irradiation and average monthly temperature.

## 2.2. Wind Turbine Model

Hourly power generation from a wind turbine can be evaluated by applying Equation (3) [7,22], which is based on a Weibull distribution and the features curve of the wind turbine [27]. $V$ is wind velocity [m/s] at the hub height of the wind turbine; $V_{cut-in}$ and $V_{cut-out}$ are cut-in and cut-out wind speed:

$$P_W(t) = \begin{cases} 0 & V < V_{cut-in} \ \ V > V_{cut-out} \\ V^3 A - P_{rate} B & V_{cut-in} < V < V_{rate} \\ P_{rate} & V_{rate} < V < V_{cut-out} \end{cases} \tag{3}$$

The constants A and B can be given by Equations (4) and (5):

$$A = \frac{P_{rate}}{V^3_{rate} - V^3_{cut-in}} \tag{4}$$

$$B = \frac{V^3_{cut-in}}{V^3_{rate} - V^3_{cut-in}} \tag{5}$$

Assuming that each wind turbine gives the same amount of electricity, and neglecting the fluid dynamic interactions, the energy produced by $N$ wind turbines ($E_{wt}$) can be calculated using Equation (6) [27]:

$$E_{WT} = N \times P_w \tag{6}$$

The hourly wind speed was obtained from [34], and depicted in Figure 3. Wind speed at wind turbine hub height can be assessed by means of Equation (7):

$$V_{hub} = V_{ref} \left( \frac{h_{hub}}{h_{ref}} \right)^u \tag{7}$$

where $V_{ref}$ is wind speed measured at reference height $h_{ref}$. $V_{hub}$ is wind velocity estimated at hub height $h_{hub}$. $\mu$ is the Hellman coefficient, which depends on ground roughness [27].

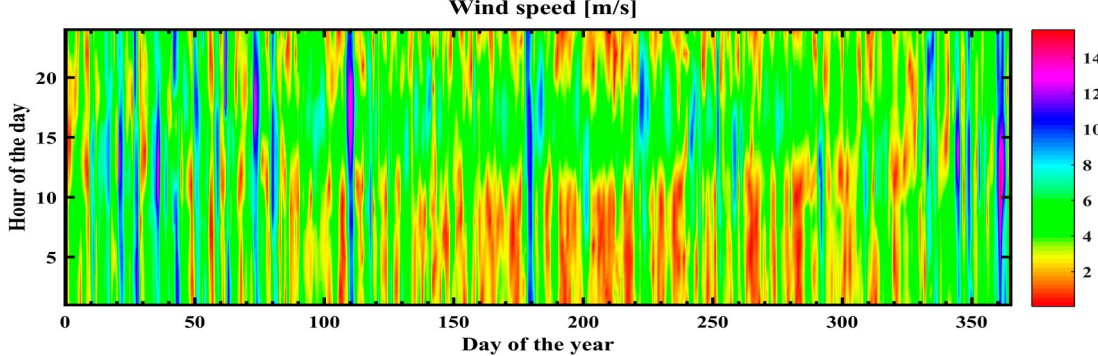

**Figure 3.** Average hourly wind speed [m/s].

### 2.3. Storage Systems

The variation of renewable energy sources (RES) can be categorized by daily and seasonal fluctuations, which are the differences in meteorological data during the day and the year, respectively [21]. Therefore, standalone hybrid PV–WT system requires a storage energy system to overcome the intermittency of variable RES and covering the load demand [7,35]. In this paper, two energy storage technologies are coupled with the PV–WT system, namely, pumped storage hydroelectricity and battery bank. A detailed description of the models applied to perform a convenient comparison between PSH and BB can be found in Appendix A.

## 3. Sizing Objectives and Optimization

As long as the installed capacity of different renewable sources of the hybrid system is a major part in its design [26], assessing the share of each source is a complicated problem [36]. The main objective of this study is to minimize the cost of electricity (COE) and loss of supply probability (LPSP) (see Equation [8]):

$$\text{fitness} = \min \begin{cases} [COE, LPSP], & LPSP > 0.2 \\ [COE, LPSP] + penalty\ factor, & \text{Otherwise} \end{cases} \tag{8}$$

To maintain the reliability of the system above 80%, a penalty factor is associated to LPSP values of more than 20%. In order to satisfy the above mentioned objectives, a powerful meta-heuristic optimization known as MOGWO is applied to solve the optimal sizing of the hybrid PV–WT–PSH and PV–WT–BB systems.

### 3.1. Objectives

The cost of electricity [€/kWh] is the result of dividing the total annualized cost (TAC) [€] by the sum of load [7,22,37].

$$COE[€/\text{kWh}] = \frac{TAC[€]}{\sum_{t=1}^{8760} E_{load}} \tag{9}$$

For the calculation of TAC for the whole system, Equation (10) given by [6,38,39] was applied.

$$TAC = \left(\sum NPC_{cp}\right) \times \frac{i(1+i)^{PLFT}}{(1+i)^{PLFT} - 1} \tag{10}$$

where *i* represents the real discount rate, and PLFT is the project lifetime.

The net present cost (NPC$_{cp}$) of each component of the hybrid system is the sum of initial cost (IC), replacement cost (RC) and maintenance and operation cost (MC), minus the salvage value earnings from its lifetime [39]. More details about the calculation of NPC$_{cp}$ can be found in Appendix B.

The reliability of the systems is indicated on the basis of LPSP [7,22,36,40] where its value is computed based on a chronological simulation technique, as shown in Equation (11).

$$LPSP[\%] = \frac{\sum_{t=1}^{8760} Eloss}{\sum_{t=1}^{8760} E_{load}} \tag{11}$$

Table 1 summarizes the techno-economic data of system components used in this study [41].

**Table 1.** Input data.

| Description | Data |
|---|---|
| Project lifespan [year] | 25 |
| Real discount rate [%] | 8 |
| **PV system** | - |
| $C_{cp}$ [€/kW] | 700 |
| $C_{PV}^{o\&m}$ [%] | 2.5% of C$_{cp}$ |
| *CLFT* | 25 |
| **Wind turbine** | - |
| $C_{cp}$ [€/kW] | 1750 |
| $C_{WT}^{o\&m}$ [%] | 2.5% of C$_{cp}$ |
| $P_{rate}$ [kW] | 225 |
| Hub height [m] | 100 |
| $V_{rate}$ [m/s] | 14 |
| $V_{cut\text{-}in}$ [m/s] | 3.5 |
| $V_{cut\text{-}out}$ [m/s] | 25 |
| *CLFT* | 25 |
| **Battery** | - |
| $C_{cp}$ [€/kWh] | 170 |
| $C_{bat}^{o\&m}$ [%] | 1.5% of C$_{cp}$ |
| $\eta_{bat}$ [%] | 85 |
| *CLFT* | 10 |
| **PSH station** | - |
| $C_{cp}$ [€/kW] | 1051 |
| Reservoir [€/kWh] | 10 |
| $C_{bat}^{o\&m}$ [%] | 1.5% of C$_{cp}$ |
| $\eta_{psh}$ [%] | 80 |
| *CLFT* | 60 |

### 3.2. Multi Objective Grey-Wolf Optimizer MOGWO

Inspired by the hunting technique and leadership of grey wolves, a new multi-objective optimization algorithm was developed by Mirjalili et al. [29], and is known as the Multi Objective Grey Wolf Optimizer. The present research is based on the adaptation of MOGWO for solving the design of a hybrid PV–WT–PSH/BB system. In this algorithm, the first step is acquisition of input data (Table 2) to generate a population of wolves that are categorized by alpha, beta, delta and omega wolves. Alpha,

beta, and delta wolves act as the leadership wolves and guide the omega wolves throughout the search space in order to select possible solutions. During the optimization process, the hunting behaviors of grey wolves are simulated and each search agent is calculated along the following formulas:

$$\vec{D}_n = \left| \vec{C}_m \cdot \vec{X}_n(t) - \vec{X}(t) \right|, \qquad n \in (\alpha; \beta; \delta), \; m \in (1; 2; 3) \tag{12}$$

$$\vec{X}_m(t) = \vec{X}_n(t) - \vec{A}_m \cdot \left( \vec{D}_n \right), \quad n \in (\alpha; \beta; \delta), \; m \in (1; 2; 3) \tag{13}$$

$$\vec{X}(t+1) = \frac{\vec{X}_1(t) + \vec{X}_2(t) + \vec{X}_3(t)}{3} \tag{14}$$

where $t$ indicates the current iteration. $\vec{A}$ and $\vec{C}$ are coefficient vectors, and can be calculated as follows:

$$\vec{A} = 2\vec{a} \cdot \vec{r}_1 - \vec{a} \tag{15}$$

$$\vec{C} = 2 \cdot \vec{r}_2 \tag{16}$$

**Table 2.** Multi-objective grey-wolf optimizer (MOGWO) input parameters.

| Parameters | Data |
|---|---|
| Number of iterations | 300 |
| Grey wolf population size | 100 |
| Archive Size | 100 |
| Repository Member Selection Pressure | 2 |
| Leader Selection Pressure Parameter | 4 |
| $\alpha$ Grid Inflation Parameter | 0.1 |
| Number of Grids per Dimension | 10 |

Over the course of iterations, elements of $\vec{a}$ decrease linearly from 2 to 0. Meanwhile, $r_1$, $r_2$ are random vectors in [0, 1]. MOGWO saves and retrieves non-dominated Pareto optimal solutions using an external fixed-sized archive. Further details can be found in [29].

*3.3. CO$_2$ Emissions*

The environmental impact of the renewables based hybrid power station has been calculated based on data from [42–45]. The emissions level was at 0.049 kg $CO_2$-eq/kWh for the PV generation, and at 0.034 kg $CO_2$-eq/kWh for wind generation. In the case of the battery storage system the emissions are connected to the storage capacity and were assumed to be 175 kg $CO_2$-eq/kWh [44]. For the pumped-storage hydroelectricity, the emissions occur both during construction (35 kg $CO_2$-eq/kWh of storage capacity) and for each unit of generated electricity, which translates to 0.0018 kg CO2-eq/kWh. The emissions for the Algerian power system were estimated based on the assumption that each unit of energy (kWh) generated translates to emissions of 0.84 kg $CO_2$-eq/kWh from an oil-powered station and 0.469 kg $CO_2$-eq/kWh from gas-powered stations [45]. Considering the fact that gas power stations cover 93% of the demand, 6% by oil and 1% by hydropower (assumed to be emissions free); the resulting average emissions in Algeria are equal to 0.518 kg $CO_2$-eq/kWh of energy.

## 4. Results

Acquisition and analysis of electrical load and the renewable energy sources availability time series is a critical step in the design of standalone hybrid PV–WT–Storage systems. In this paper we investigated, in total, eight hourly load time series covering a single year. They represented different electricity customers such as a university, a hospital, a farm and various industries. A ninth

load was created by an aggregation of all eight loads and is intended to represent an ensemble of electricity consumers. For each load the same procedure was applied, which can be summarized in the following steps:

1. Data acquisition and processing (screening for inconsistencies, such as out-of-scale values).
2. Searching for an optimal system configuration (both PV–WT–PSH and PV–WT–BB) based on the methods and models presented in Section 3. Selecting system parameters that represent the optimal solution based on the Pareto front as well as those that enable the system to obtain a reliability of 95% (arbitrarily selected).
3. Performing a sensitivity analysis with regard to the system reliability based on the historical time series of wind speed and irradiation covering the last 10 years.
4. Estimating the system's environmental impact in terms of $CO_2$ emissions per unit of covered energy demand.

In the following subsections each load will be briefly characterized and the results of system operation will be presented. The outcome of the sensitivity analysis and further implications of our research will be presented in the discussion section.

### 4.1. Case Study 1: Farm Load

The first considered load represents the electricity consumption of a farmstead. The largest portion of consumption results from irrigation of more than 20 hectares of fruit plants. The annual demand amounts to 0.35 GWh and is characterized by a clear annual pattern (Figure 4) where the highest consumption is observed from May to late October. The hourly demand ranges from 0.0 to 193.5 kWh, with a mean value of 40.1 kWh. Notably, the majority of the load is observed from the early morning hours to the late afternoon—which might be beneficial from the perspective of PV system operation.

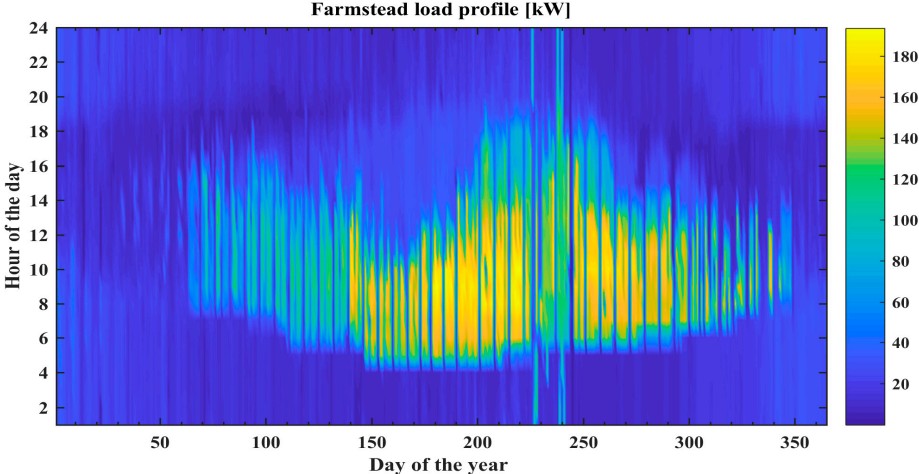

**Figure 4.** Spectrum of electricity load demand in farmstead.

The computational results indicate that the corresponding *COE* for a reliability of 95% is 0.135 €/kWh for a PV–WT–PSH system and 0.286 €/kWh for a hybrid with BB. In both systems, wind energy plays a relatively small role, and its installed capacity is 36 kW (PSH system) and 2.25 kW (battery system), as compared to 334 kW and 451 kW for PV. The storage capacity is relatively high and the PV–WT–PSH has a storage capacity of 4.8 MWh (208.5 kW of pumping/generating capacity), whereas the system with battery is capable of storing 2.1 MWh of electricity, which amounts to about two days of energetic autonomy. In the case of the farmstead, the system with PSH storage is not only superior to the battery system in terms of electricity cost but also its environmental impact is smaller. Covering load from the PV–WT–PSH system will result in emissions of 0.064 kg $CO_2$-eq/kWh, as opposed to 0.364 kg $CO_2$-eq/kWh from the battery system. Reliability of 100% was only obtained for

the battery system (due to the constraint on the minimal operating load of the PSH). However, the cost of reliability is very significant, and, in the case of the battery system, the *COE* increased to 0.443 €/kWh, and for the system with PSH (whose maximal reliability was 98.5%) amounted to 2.58 €/kWh.

### 4.2. Case Study 2: Hospital

The hospital's is one of the critical loads which is vital for the successful operation of the whole municipality. Providing electricity to hospitals and maintaining the maximum possible level of reliability is of the highest importance. The hospital considered in this case study employs an average of 282 physicians and has a capacity of 120 beds. The hourly electricity demand (Figure 5) ranges from 26.3 kWh to 191.5 kWh with a mean of 78.5 kWh. The resulting annual consumption of 0.68 GWh also has a certain seasonality. The highest demand is observed from late November until the end of February and later from late June to early August (Figure 5). This load is also characterized by a relatively low variability. Its coefficient of variation (standard deviation divided by mean) is only 33%, whereas for the farmstead it was close to 120%.

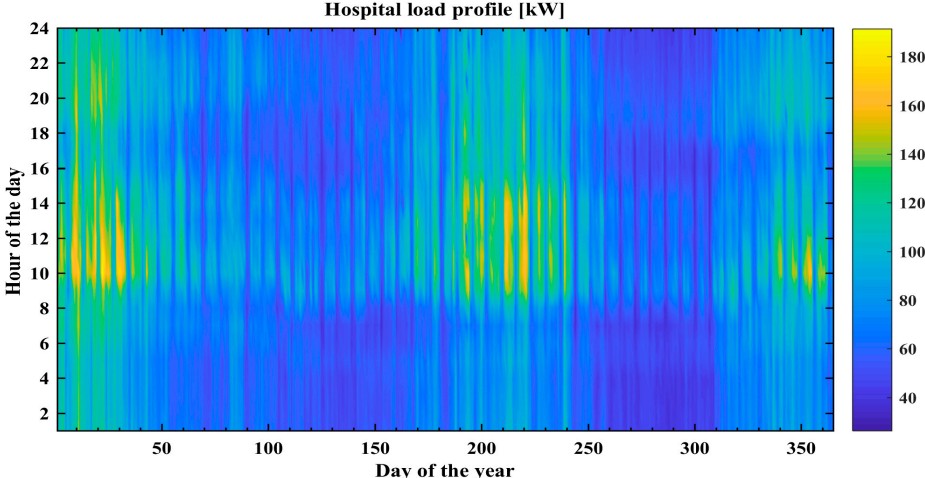

**Figure 5.** Spectrum of electricity load demand of hospital.

The performed optimization indicates that 95% reliability can be achieved at a *COE* of 0.088 €/kWh for the hybrid PV–WT with PSH and 0.238 €/kWh for the system with battery. Similarly as with the farmstead load, the majority of the capacity will be installed in PV systems—respectively, 0.52 MW (PSH) and 1.06 MW (battery). The capacity of wind generation in the case of the hybrid with PSH is 0 kW and for the battery system does not exceed 80 kW. Again the storage capacity of the PV–WT–PSH system is much greater and amounts to 11.1 MWh, in contrast to only 2.4 MWh for the battery system. The emissions from the PV–WT–PSH system are similar to the farmstead case study and amount to 0.063 kg $CO_2$-eq/kWh, whereas for the battery system it is 0.23 kg $CO_2$-eq/kWh. Naturally, as mentioned at the beginning of this subsection, it is of extreme importance to ensure very high reliability of such supply systems. In both cases, it was possible to obtain 100% reliability, although the resulting *COE* was high, amounting to 2.06 €/kWh for the PSH system and 0.471 €/kWh for the battery system.

### 4.3. Case Study 3: University

The third load was obtained from the University of Mostaganem. The mentioned university employs over 830 teaching staff and educates some 28,362 students. For this load a clear daily energy consumption pattern can be observed (the majority of energy is used from late morning hours until 4–5 PM) (Figure 6). This situation is again very beneficial with regard to energy produced from PV systems. The observed electricity demand ranged from 1.5 kWh to 204 kWh, with a mean hourly demand of 34.8 kWh. This load, similarly as in the case of the farmstead, is characterized by relatively high variability (CV = 93%).

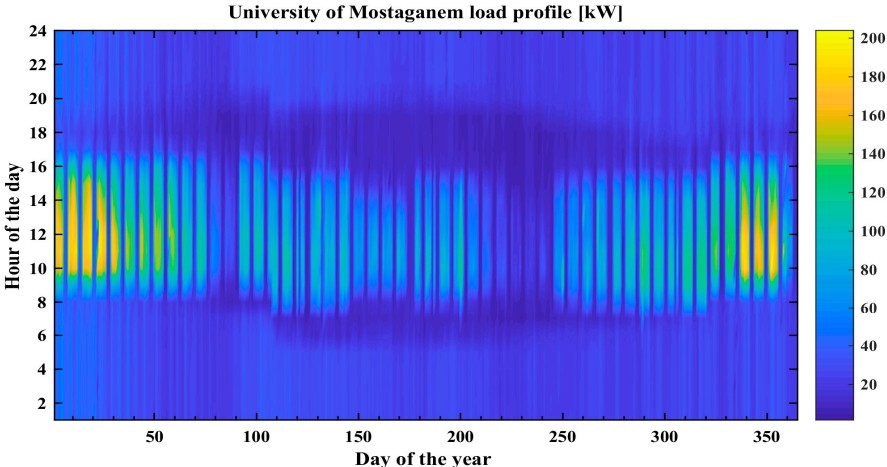

**Figure 6.** Spectrum of electricity load demand of the University of Mostaganem.

Based on the performed optimization it was found that, in the case of the PV–WT–PSH system, the load can be met at 95% reliability. The hybrid system will consist of 126 kW of PV, 72 kW installed in wind generation, 119 kW of pumping/generating capacity of PSH with a storage capacity of 10.32 MWh (more than 12 days' autonomy). For the second system, 510 kW in PV, 27 kW in wind generation and 1.83 MWh in battery storage ensured 95% reliability. Again, the cost of the PSH system was lower and resulted in a *COE* of 0.128 €/kWh, and the battery system was almost three times more expensive in terms of *COE* (0.325 €/kWh). The emissions from the PSH systems were six times lower than from the battery system (respectively 0.056 and 0.364 kg $CO_2$-eq/kWh). The 100% reliability was only achieved in the case of the battery system; however, it almost doubled the electricity price (*COE* = 0.643 €/kWh), whereas the most reliable PSH system yielded an electricity price of 0.8 €/kWh.

### 4.4. Case Study 4: Hotel Load

The next considered load is representative for a 150-room hotel. The electricity demand exhibits a clear peak demand during summer (from late June to early September) when the hotel operates at full capacity (Figure 7). The overall energy demand on an annual scale amounts to 0.4 GWh and has a mean hourly electricity consumption of 46 kWh. The hourly demand ranges from zero kWh to almost 200 kWh. The daily demand pattern is less distinguishable. However, greater peak demand is observed during the night than the day, which may imply that energy surpluses from the PV operation should be stored to be used during the night.

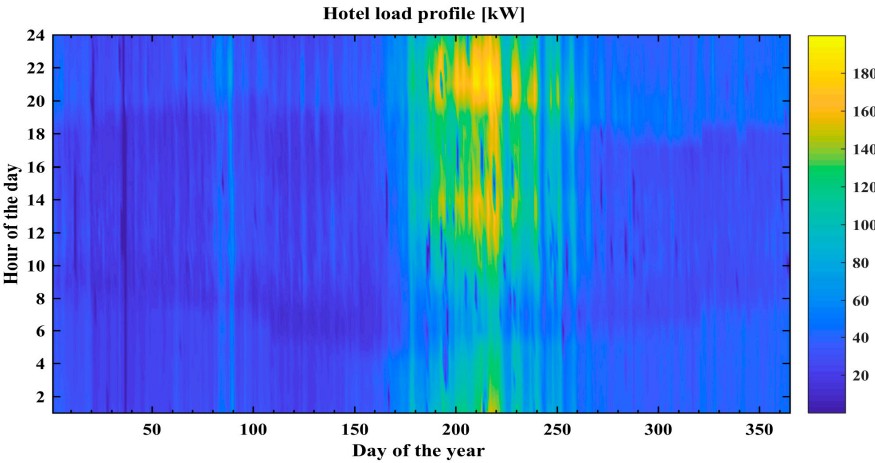

**Figure 7.** Spectrum of electricity load demand in hotel.

From the optimization viewpoint, it is found that in the case of both systems (and desired reliabilities of 95% and maximal possible) there is no justification for using wind energy, meaning that both will be powered only by PV generation. For the system with PSH as storage, the lowest possible price ensuring reliability at 95% was found to be 0.14 €/kWh, whereas for the system with the battery it was 0.24 €/kWh. The PV–PSH should have a capacity of PV equal to 0.6 MW, 178.7 kW of turbine/pump capacity and a storage potential of 5.6 MWh. In the case of the battery system, reductions in both PV capacity (now 0.57 MW) and storage capacity (now 1.8 MWh) was observed. The emissions from both systems are comparable to the previous case studies, and amount to 0.063 and 0.285 kg $CO_2$-eq/kWh for PSH and battery system, respectively. 100% reliability was only achieved for the battery system, but it results in doubling *COE* (0.49 €/kWh) and significant system oversizing. In the case of the PSH system, the maximal achieved reliability was 99.5%, but it increased the energy price almost nine-fold and resulted in overall oversizing of storage, a reduction in pump/turbine capacity, and a PV capacity of close to 7 MW.

### 4.5. Case Study 5: Industrial Load—Brickyard

The first industrial load is one of the biggest ones considered in terms of annual electricity consumption, which in this case amounts to 5.34 GWh. The load is representative for a brick manufacturing facility. The observed hourly demand (Figure 8) ranges from 0 kWh to almost 1160 kWh, with a mean value of 610 kW. Simultaneously this load (compared to the remaining ones) is characterized by a relatively low variability (CV = 34%). The load does not exhibit any significant daily or seasonal changes in the electricity consumption (Figure 8). The average increase in electricity consumption after day 225 may be connected to increased production in response to market operation.

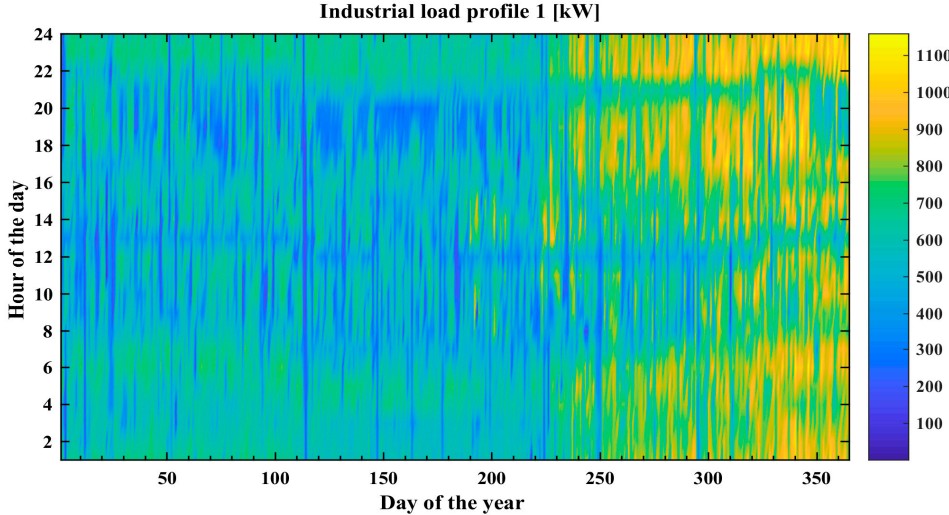

**Figure 8.** Spectrum of electricity demand of brick manufacturing facility.

After performing the simulation it was found that 95% reliability can be ensured by PSH and battery systems at respective costs of 0.08 and 0.18 €/kWh. In both systems a negligible role is played by wind generation, at 11 kW (PSH) and 168 kW (battery). In the case of the PV–WT–PSH system, the PV capacity reaches 3.8 MW and the storage capacity is equal to 65.8 MWh, with a pumping/generating capacity of 1.93 MW. The PV–WT–battery system has an oversized PV generation of 6.48 MW, with a significantly smaller storage potential of close to 16 MWh. Again the PSH-based system has lower emissions of 0.06 kg $CO_2$-eq/kWh, compared to 0.2 kg $CO_2$-eq/kWh from the battery system. 100% reliability was achieved for the battery-based system (at a *COE* of 0.54 €/kWh) whereas the PV–WT–PSH system resulted in a reliability of 99.95% for a slightly higher cost of 0.6 €/kWh.

### 4.6. Case Study 6: Industrial—Grain Mill

The second analyzed industrial load (Figure 9) represents a grain mill. This mill employs some 50 employees and produces wheat flour. The mill's operation shows distinct daily and weekly energy demand patterns. Also, the mill stops operating in the early evening hours and starts again after 10 PM. During the considered period, the mill operation was stopped several times for shorter and longer periods (see, for example, days 120–160 in Figure 9) The annual electricity demand amounts to 0.75 GWh and, due to frequent periods of no operation, has quite big variability (CV = 99%). The maximal observed load was 231.7 kWh, the lowest was zero kWh, and the mean hourly demand was 86.5 kWh.

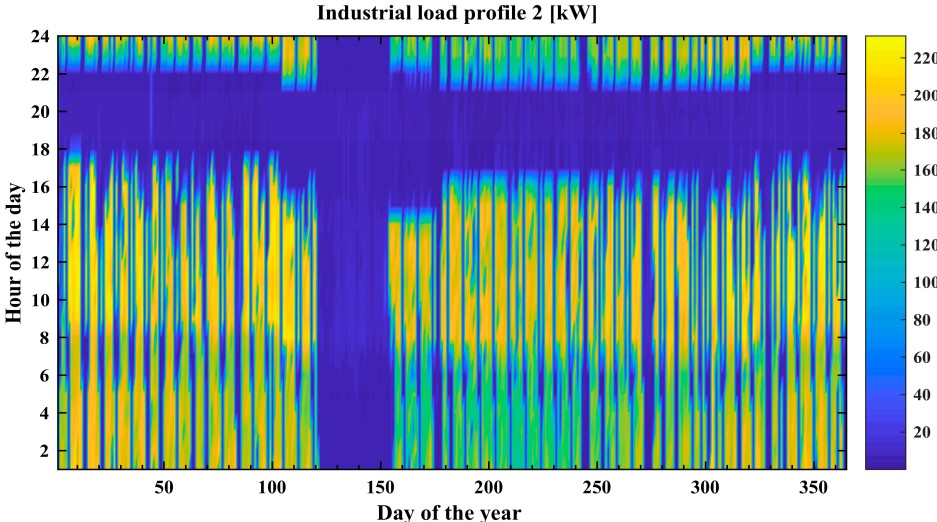

**Figure 9.** Spectrum of electricity demand of grain mill.

After executing the optimization the results indicate that 95% reliability can be achieved for *COE* = 0.08 €/kWh for a PSH system and, in the case of the battery system, for almost three times more (COE = 0.23 €/kWh). In case of this load the battery system consists of 1.1 MW in PV, 6.7 kW in wind generation and storage capacity of 3.0 MWh. The PSH-based system has much smaller capacity of PV generation (0.5 MW) but has much greater storage capacity, at 14.7 MWh with a pumping/generating capacity of 238 kW. The PV–PSH system again has much lower emissions of 0.06 kg $CO_2$-eq/kWh, whereas the battery system's is almost four times higher (0.23 kg $CO_2$-eq/kWh). 100% reliability was achieved only by the battery system, but resulted in a doubling of electricity cost (*COE* = 0.46 €/kWh), whereas for the PSH system the achieved reliability of 99.25% increased the cost over 12-fold, to 0.96 €/kWh.

### 4.7. Case Study 7: Industrial Load—Water Pumping Station

The third industrial load represents a crucial part of the municipal subsystem in the form of a water pumping station. The station consists of four pumps of 1 MW capacity each. In terms of annual energy consumption, this load (31.3 GWh) is the biggest considered in this paper. The average hourly demand amounts to 860 kWh, and over the year ranges from 10 kW to over 4.2 MW. However, this load also has the lowest overall variability (CV = 24%). This load has no substantial seasonal energy demand patterns, although it can be observed (Figure 10) that the pumping station usually operates under lower loads during the late night hours.

After performing the optimization it was found that, once again, the PSH-based hybrid system was superior in terms of providing a lower electricity price for a reliability of 95%. A PV–WT–PSH system using 31.7 MW in PV, 191 kW in wind generation, almost 6 MW in pump/turbine and 468 MWh of storage capacity delivers electricity at *COE* = 0.093 €/kWh. By contrast, the second, battery-based

system, has an installed capacity in PV of 32.2 MW, 1.19 MW in wind generation and 82.2 MWh in storage and yields *COE* = 0.16 €/kWh.

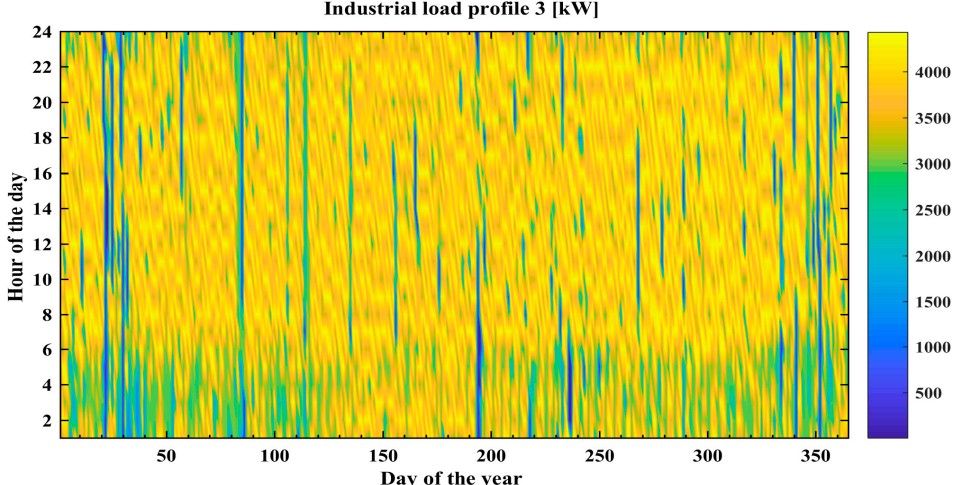

**Figure 10.** Spectrum of electricity demand of water pumping station.

Again the emissions are generally lower for the PSH system (0.061 kg $CO_2$-eq/kWh), but a decrease in emissions (comparing to the loads already presented and those remaining) was also observed for the battery-based system (0.188 kg $CO_2$-eq/kWh). 100% reliability was almost achieved for both systems (99.94% for the PSH and 99.99% for the battery system) but it results in an energy cost increase to 0.13 €/kWh for the PSH and 0.42 €/kWh for the battery-based system. Interestingly, a lower relative price increase was observed for the PSH system, indicating its overall superiority in covering larger electrical loads.

### 4.8. Case Study 8: Industrial—Food Factory

The last load is also the second in terms of annual electricity consumption (18.3 GWh). It represent a load characteristic for the manufacturing processing of food like pasta. The considered load (Figure 11) does not exhibit any significant seasonal or daily patterns of electricity consumption. The maximal observed energy demand reaches 3.0 MW and the lowest 0.3 kW. Despite the relatively large range of electricity demand the load has quite low variability (CV = 26%, also one of the lowest of those considered). The mean hourly demand is 2.07 MWh.

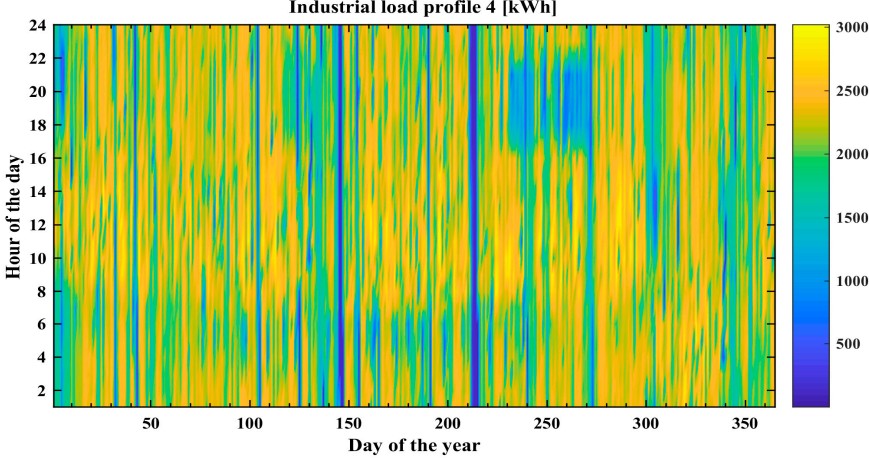

**Figure 11.** Spectrum of electricity demand of food processing facility.

The optimization revealed (Figure 12) that for 95% reliability the PV–WT–PSH system can guarantee the lowest cost of electricity (*COE* = 0.07 €/kWh) which is also the lowest of all considered loads. The COE of the battery system is significantly higher and amounts to 0.17 €/kWh. The configuration of the PSH-based system should include 12.2 MW in PV generation, a mere 13.5 kW in wind generation, a pumping/generating capacity of 5.36 MW and a significant 188.8 MWh of storage capacity. The battery system has a lower storage capacity (50.2 MWh) but a higher generation capacity; 20.5 MW for PV and 0.6 MW for wind turbines. The emissions from the PSH-based system are three times smaller than from the battery system, at 0.62 and 0.19 kg CO2-eq/kWh, respectively. However, none of the systems was able to ensure reliability at a 100% level. The highest reliability was achieved by the battery system (99.99%), although it resulted in general oversizing and a doubling of the COE (0.35 €/kWh). For the PSH-based system the electricity price increased less than two-fold, to 0.12 €/kWh, with a reliability just over 0.1% worse than for the battery system.

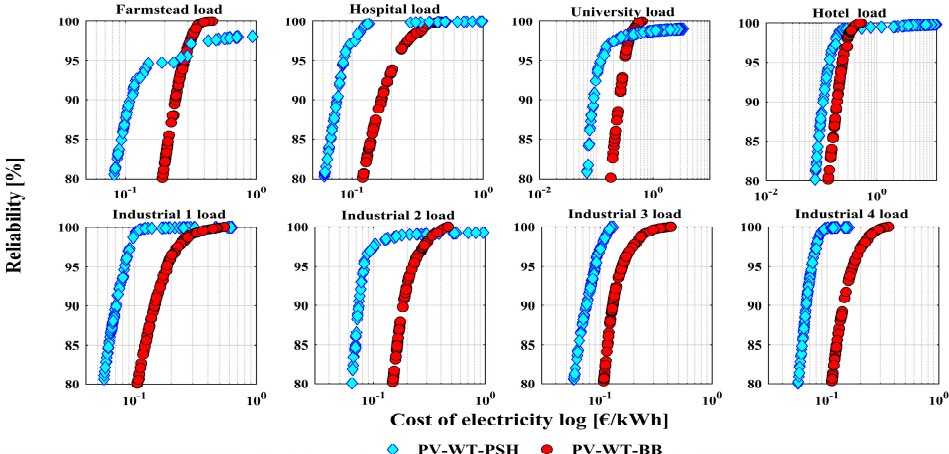

**Figure 12.** Pareto fronts of optimal solutions selected by Multi-objective grey wolf optimizer (MOGWO).

## 5. Discussion

This section is divided into five subsections, which discuss, in turn: the parameters and operation of hybrid systems serving the aggregated load; the economic performance of hybrid systems comparing to current electricity prices in Algeria; the environmental impact of the hybrid system in comparison to the emissions from the Algerian power system; system reliability; future research directions and study limitations.

### 5.1. Aggregated Load

The final optimization was executed for the aggregated load (based on the earlier considered eight case studies). The annual sum of this load (Figure 13) is 57.3 GWh. The aggregation of loads results in a decrease in demand variability (CV = 16%). The highest demand reaches 8.63 MWh and the lowest 1.29 MWh. The resulting hourly mean amounts to 6.55 MWh. The typical (averaged over the whole year) daily energy demand profile is presented in Figure 14. It exhibits a peak demand during daylight hours (which is very beneficial for increasing the onsite energy consumption from PV systems, thereby reducing the need for energy storage. However, a second peak demand can be seen during the late evening hours of 9–11 PM. Therefore, it is clear that this demand has to be covered either from wind generation or from storage.

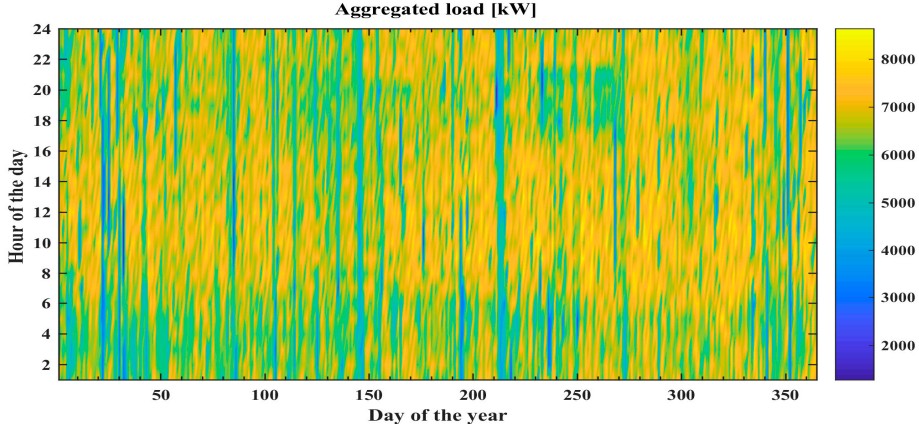

**Figure 13.** Spectrum of aggregated load's electricity demand.

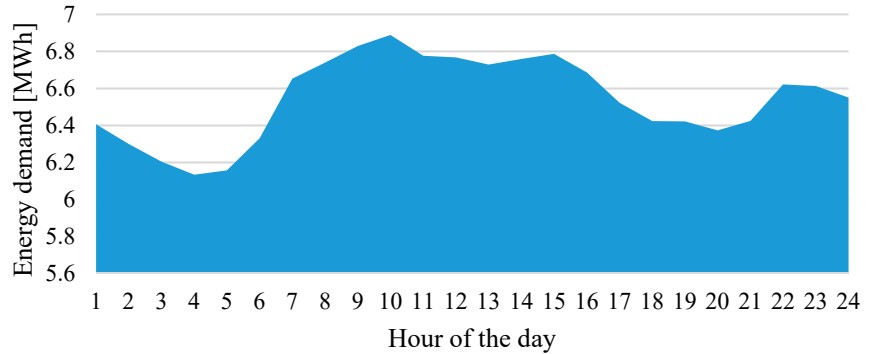

**Figure 14.** Typical daily energy demand pattern.

The optimization results (Figure 15) indicate that both systems can ensure a reliability at 95% for a *COE* of 0.07 €/kWh and 0.12 €/kWh for PV–WT–PSH and PV–WT–battery systems, respectively. A reliability of 100% can be achieved only for the battery-based system but increases the COE to 0.17 €/kWh. The PV–WT–PSH system can achieve a maximal reliability of 99.9%, but its cost will be much higher, at 0.39 €/kWh. For the systems with 95% reliability it was found that the emissions can be as low as 0.06 kg $CO_2$-eq/kWh for the PSH system and 0.14 kg $CO_2$-eq/kWh for the battery system. For the former, to achieve such economic, environmental and reliability performance it would be mandatory to install 38 MW in PV, 0 MW in wind generation, 14.8 MW in pumping/generating capacity and 393 MWh of reservoir storage. The battery system has a greater cumulative capacity of renewable generators—respectively, 36.9 MW in PV and 4.6 MW in wind generator. The storage capacity of the PV/WT–battery system is almost four times smaller, and amounts to 105 MWh.

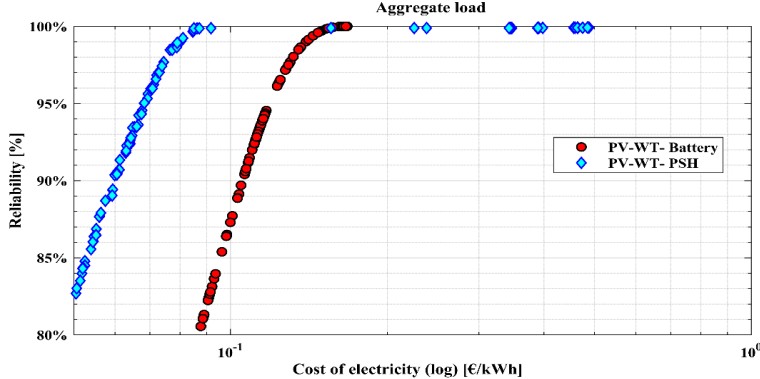

**Figure 15.** Scatter plot of Pareto front of bi-objective optimization for aggregated load.

### 5.2. Economic Performance

The current electricity prices in Algeria are relatively low when compared to, for example, the European Union. The average electricity price in Algeria is 0.035 €/kWh. This is substantially lower than the electricity cost that has been obtained here for renewables-based systems—even those with a 95% reliability (leaving aside the systems aiming to cover the whole load). Figure 16 illustrates how the electricity price for individual loads and systems relates to the mean Algerian electricity price. The cost of electricity from the renewables-based systems is on average 2.8 times higher than the electricity from the national grid for the PSH, and 6.2 times higher for the batteries-based systems. However, in the case of the aggregated load the electricity price from the PV–WT–PSH systems is greater by only 94%, whereas for the batteries-based system it is 234% higher. A potential cost reduction can be achieved by hybridizing the operation of battery and PSH storage (discussed later in Section 4.5), and more efficient utilization of the curtailed energy. Also, and this is important to highlight, this study used only the average prices for the components from [41]. Further analysis would be required to explore how the electricity cost would vary considering different values of economic input parameters. However, one thing is almost certain: the investment in a RES-based system is made only at the beginning, while the operation and maintenance costs are relatively low. Meanwhile, in the Algerian power system it is quite likely that electricity prices will increase in future, thereby making RES-powered hybrids economically superior.

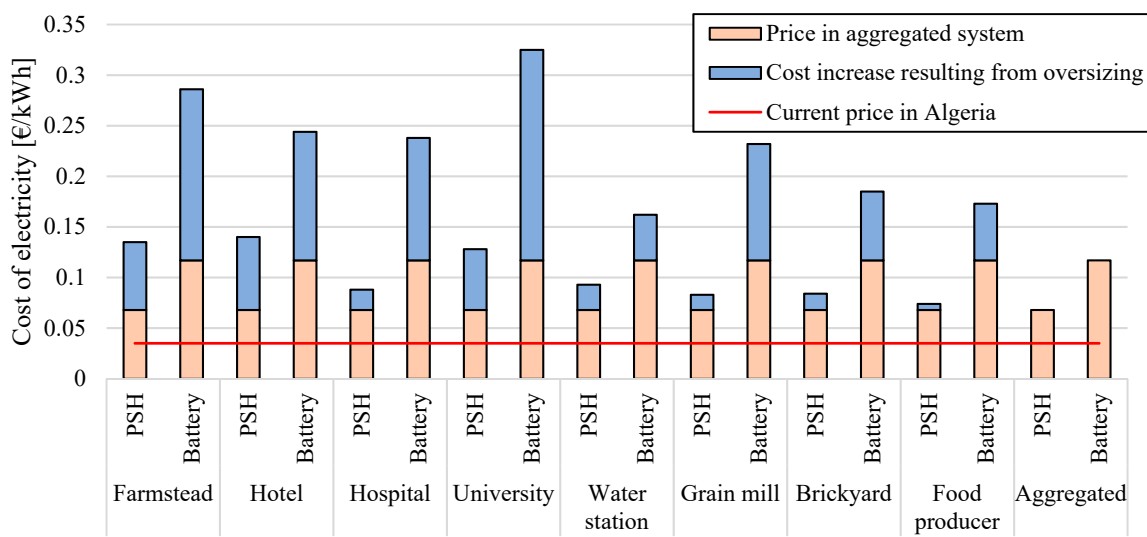

**Figure 16.** Electricity cost.

### 5.3. Environmental Impact

Renewables-based energy systems are believed to be environmentally friendly. This is not entirely true, as, in the majority of cases, conventional fuels are used for their production (at least during the initial phases of the transition to RES-dominated power systems). Therefore, each unit of generated energy is associated with some emissions appearing at the manufacturing phases. In our analysis, we estimated the energy emissions from both PSH- and battery-based system. The $CO_2$ emissions from the Algerian power system (which is dominated by oil- and gas-powered power stations) can be estimated at 0.518 kg $CO_2$-eq/kWh [45].

Figure 17 illustrates the emission from individual systems serving different loads with 95% reliability. The average emissions from the PSH-based systems are 0.055 kg $CO_2$-eq/kWh, whereas for battery systems they are 0.220 kg $CO_2$-eq/kWh, although the latter can be as high as 0.364 kg $CO_2$-eq/kWh. The lowest emissions have been observed for the aggregated load, both for PSH and battery systems. However, most importantly, those emissions are 9.3 (PSH) and 2.3 (battery) times lower than in the Algerian power system. This indicates that a clear governmental policy aiming at

promoting low-emission power systems (like taxation of conventional fuel generators) would overcome the problem with the currently higher prices from RES-based systems (as shown in Section 4.2).

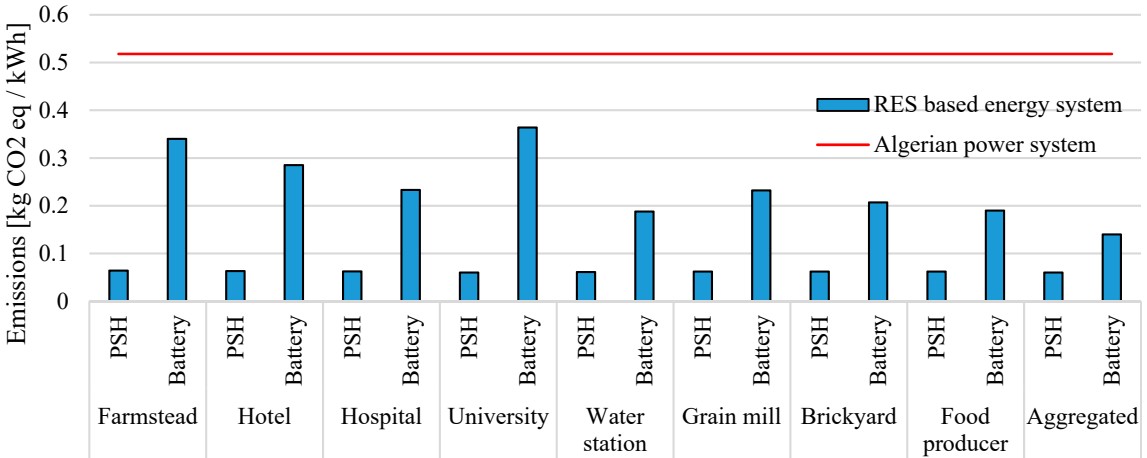

**Figure 17.** $CO_2$ emissions from renewables-based energy systems. Pumped-storage hydroelectricity (PSH) and Battery indicate type of storage. System parameters as in Section 3.

## *5.4. Reliability Analysis*

To verify the overall hybrid systems reliability where LPSP has been used as an index, we tested their performance on historical irradiation and wind speed time series that are representative for the considered location (Figure 18). However, these data were not used during the optimization of the system parameters; for that purpose, we used data covering a total of 10 calendar years. The systems parameters were as described in Section 3 for 95% reliability. Only for the hospital (which is a critical load) did we decide to test the systems which yielded a reliability of or close to 100% (PV–WT–PSH). As can be observed, for all loads the designed hybrid systems turned out to be relatively resilient. The variation from the designed reliability in most cases did not exceed 2% and in most the systems performed even better than during the "designing year". In general, the battery systems exhibited a greater variability in reliability (compared to PV–WT–PSH systems). Only in the case of the university was the opposite observed, although even there the greater reliability variability for the PSH-based system had minimal negative impact, as reliability was greater than 95%. In the case of the hospital load, it was found that reliability at 99.9% is almost certain, but it must be remembered that such reliability comes at a much greater cost.

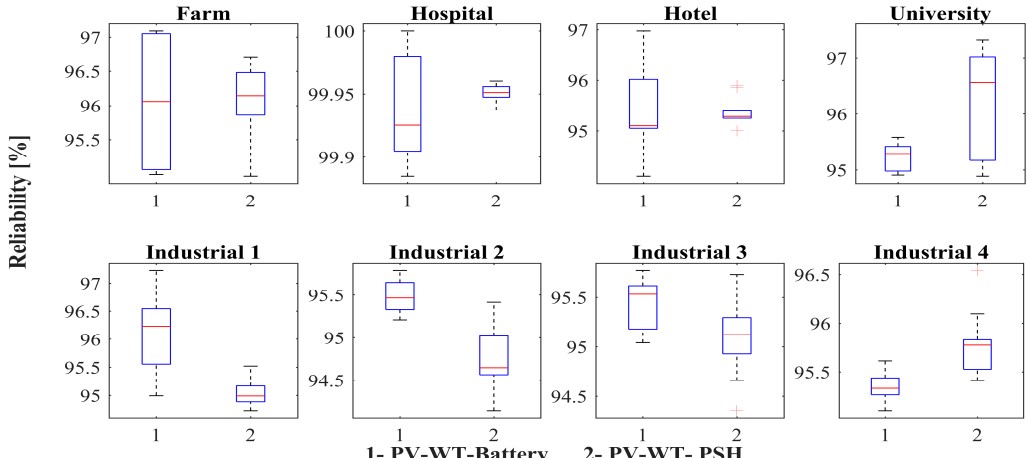

**Figure 18.** Reliability analysis for all systems based on 10 years of historical data of meteorological parameters.

As shown in Section 5.1 it is easier to supply an aggregated load from variable renewable energy-based hybrid sources. Therefore, considering the fact that the hospital load constitutes a very small part of the total load (0.6%) it can be said that by adequately prioritizing the order in which each load is served the hospital load can always be covered and human life will not be threatened. Of course, prioritizing the load is not the only solution, as load shifting can also be applied, as discussed in the next section.

### 5.5. Load Variability—Energy Cost

The final part of our analysis investigates the relationship between the load varibility (expressed in the coefficient of variation) and the cost of energy resulting from optimal system parameters for 95% reliability. Figure 19 visualizes this relation for both PV–WT–PSH and PV–WT–BB systems. As can be observed, the general tendency is that with increasing variation in load the renewables-based systems start to provide electricity at a higher cost. In those terms the hybrids utilizing PSH as a storage medium seem generally to be not only cheaper but also more resilient to increased load variability (the fitted linear trend has the value of the slope coefficient).

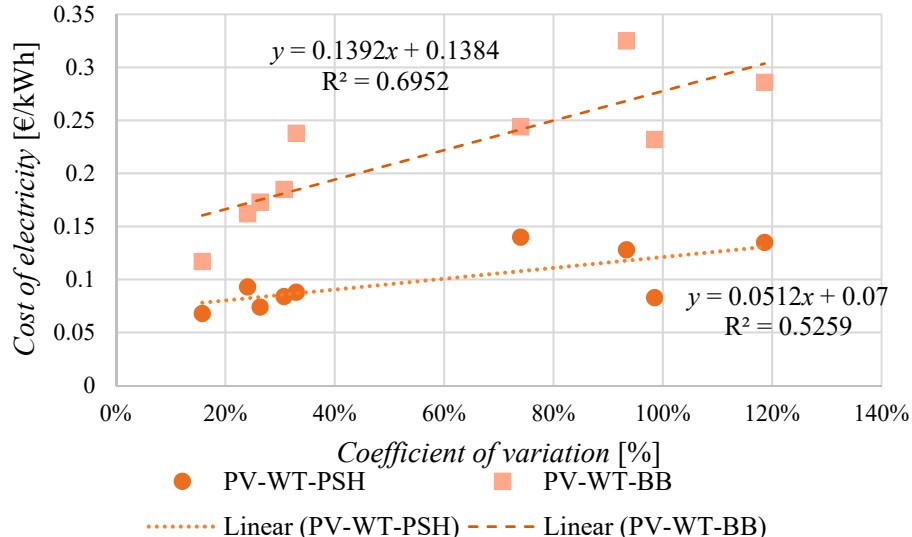

**Figure 19.** *Coefficient of variation* vs. *cost of energy* for systems with 95% reliablity.

The fact that the lowest CV value (which is characteristic for the aggregated load presented in Section 5.1) obtained the lowest cost of electricity, it can be said that the overall renewables-based power station performance may benefit from load aggregation. This also indicates that the statistical parameters of the load (such as CV) may be useful in the initial assessment of renewables-based system performance.

### 5.6. Study Limitations and Future Research Directions

The following points briefly illustrate the limitations of this study and thereby highlight the potential further research directions:

- Considering that for all case studies relatively acceptable energy prices have been observed for a reliability of 95% it is worth investigating the potential extent of the demand-side management/load shifting that would result in increased system reliability. If the load shifting could exceed 5% of the demand it might be possible to even further reduce the energy cost or, in other words, avoid oversizing the system. Also, to increase the match between supply and demand it would be interesting to investigate the different orientations of PV systems [46].
- Systems operating in off-grid mode have to ensure the required reliability, but also provide energy at an acceptable cost. The curtailment of renewables generation leads to an increased energy cost

but might be unavoidable due to the storage capacity constraints. Therefore, a tradeoff has to be made between cost and reliability. Our research shows that covering aggregated loads based on the renewable energy sources hybrid systems is more effective. However, future research should consider the use of potentially curtailed energy for the production of, for example, synthetic fuels or hydrogen, as in [47].

- The performed analysis clearly indicates that, due to the technical limitations of battery and PSH storage systems, it is necessary to in the future consider the potential option of the joint operation of those two storage technologies. Such hybridization would result in a more efficient operation of the whole system through the synergetic effect of complementary technologies [48].

- In this research, hybrid systems were operating in an off-grid mode, despite being relatively close to the national grid. In future research, it would be necessary to investigate how such a hybrid system could cooperate with the national power system by, for example, offering peak load shaving services and simultaneously minimizing the electricity cost [49].

## 6. Conclusions

The objective of this research was to investigate the potential of renewable energy sources to cover the energy demand of various electricity customers located in close proximity to each other. Our analysis has shown that the performance of variable-renewable-energy-based hybrid systems depends on the energy-demand statistical parameters (mean, range, standard deviation). Based on load measurements from a group of existing electricity consumers we provide some evidence into the discussion on the optimal sizing of hybrid renewable energy systems. Although standalone hybrid systems can ensure high reliability, this comes with a prohibitive energy cost. This results mainly from technical constraints of the storage system (pumped storage). Additionally, as a part of the discussion, the sensitivity analysis reveals that the hybrid system performance may benefit from load aggregation. Moreover, the systems are relatively resilient to changing meteorological conditions (variability of wind speed and irradiation). Regarding the Algerian case study, the renewables-based energy generation is currently superior in terms of environment protection but fails to deliver electricity at a price that is cost-competitive with the national power system.

**Author Contributions:** All authors discussed and agreed on the study design. M.G. and J.J. formulated the optimization problems, M.G. aggregated the input data and performed the optimization, J.J. and M.G. wrote the paper and discussed the obtained results, and B.B. supervised the research, reviewed the final version and provided the results validation.

**Funding:** This research received no external funding.

**Acknowledgments:** M.G. wishes to acknowledge Robaï Samir for providing the loads' data.

**Conflicts of Interest:** The authors declare no conflict of interest.

## Nomenclature

| | |
|---|---|
| $C_{cp}^{o\&m}$ | Maintenance and operation cost of component (€) |
| $C_{cp}^{R}$ | Replacement cost of component (€/KW) |
| $C_{slv}$ | Salvage value (€) |
| $E_b$ | Stat of charge of Battery (kWh) |
| $E_{psh}^{G}$ | Energy generated from PSH station |
| $E_{psh}^{P}$ | Energy pumped to upper reservoir |
| $NPC_{cp}$ | Net present cost of component (€) |
| $N_{rep}$ | Number of replacements of component |
| $P_{cp}$ | Installed capacity of component (KW) |
| $E_{Load}$ | Load demand (kWh) |
| $P_{Rs}$ | Energy produced by renewable sources (kWh) |
| BB | Battery Bank |
| CLFT | Component lifetime (year) |

| | |
|---|---|
| COE | Cost of energy (€/kWh) |
| $E_{dump}$ | Curtailed energy |
| Eloss | Uncovered energy |
| evap | Evaporation (m³) |
| h | Head (m) |
| i | Real discount rate |
| IC | Initial cost (€) |
| LPSP | Loss of supply probability |
| MC | Maintenance and operation cost (€) |
| MOGWO | Multi-objective grey-wolf optimizer |
| $\eta_{Bat}$ | Battery efficiency |
| $\eta_{psh}$ | Water turbine efficiency |
| $P_{dif}$ | Deficit power |
| PLFT | Project lifespan (years) |
| PSH | Pump storage hydroelectricity |
| $P_{ss}$ | Surplus enegy |
| PV | Photovoltaic energy |
| Qpsh | Water turbine throughput (m³/s) |
| $rain_{fall}$ | Precipitation (m³) |
| RC | Replacement cost (€) |
| RES | Renewable energy system |
| TAC | Total annualized cost (€) |
| V | Volume of water (m³) |
| WT | Wind energy |
| $\rho$ | Water density (Kg/m³) |

## Appendix A.

### Appendix A.1. PSH Operation

The pumped hydro-storage system simply exploits the excess energy from RES to pump water from a lower reservoir (sea) to an upper reservoir [50]. The operation of PSH is modeled by the following equations: Equations (A1)–(A7) and those are given by [11,50]:

$$P_{psh} = \rho\, Q\, g\, h\, \eta_{psh} \tag{A1}$$

where $P_{psh}$ is the nominal capacity of the water turbine (Kw), $Q$—water flow rate (m³/s), $g$—gravitational acceleration 9.8 (m·s$^{-2}$) and $h$—hydropower head 380 (m). $\eta_{psh}$ is PSH efficiency and can be estimated at 80% in both charge and discharge mode [21]:

$$E_{psh} = \left(\frac{V_{(t)}}{3600[s]}\right)\rho\, g\, \eta_{psh}\left(\frac{V_{(t)}}{s} + h\right) \tag{A2}$$

where $E_{psh}$ is the energy stored in the upper reservoir (kWh), $V_{(t)}$—the volume of water stored at instant $t$ (hour) and $s$—the surface of the upper reservoir (m²). This latter can be evaluated using Equation (A3):

$$V_{(t)} = V_{(t-1)} + rain_{fall(t-1)} + V_t^{Dis} - evap_{(t-1)} - V_t^{Pump} \tag{A3}$$

where $rain_{fall(t-1)}$ and $evap_{(t-1)}$ are precipitation and evaporation from the upper reservoir, and the data was retrieved from [33].

The volume of water ($V_t^{Dis}$) used to generate electricity ($E_{psh}^G$) can be calculated by Equations (4) and (5):

$$E_{psh}^G = \min\left[\min\left(\frac{V_{t-1}}{3600[s]}; P_{psh}\right)\rho\, g\, \eta_{psh}\left(\frac{V_{t-1}}{s} + h\right); P_{dif}\right] \tag{A4}$$

$$V_t^{Dis} = \frac{E_{psh}^G}{\rho \, g \, \eta_{psh} \left( \frac{V_{t-1}}{s} + h \right)} \tag{A5}$$

The volume of water $\left( V_t^{Pump} \right)$ pumped into the upper reservoir is calculated based on the following equations:

$$E_{psh}^P = \min \left[ \min \left( \frac{V_{max} - V_{t-1}}{3600[s]} ; P_{psh} \right) \rho \, g \, \eta_{psh} \left( \frac{V_{t-1}}{s} + h \right) ; P_{ss} \right] \tag{A6}$$

$$V_t^{Pump} = \frac{E_{psh}^P}{\rho \, g \, \eta_{psh} \left( \frac{V_{t-1}}{s} + h \right)} \tag{A7}$$

The following flowchart (Figure A1) describes the energy management and the operation of PSH in both cases of pumping and generation, where the operation of PSH can fairly be assumed to be constant when keeping between 15% and 100% of its nominal capacity (Qpsh).

*Appendix A.2. Battery Bank Modeling*

The amount of energy $E_b$ *(t)* (kWh) stored in a battery bank at hour (*t*) generally depends on the overall energy $P_{Rs}$ (kW) produced from hybrid PV–WT sources and can be calculated as follows [7,51]:

$$E_b(t) = E_b(t-1) + [P_{Rs}(t) - P_{Load}(t)/\eta_{inv}] \times \eta_{Bat} \tag{A8}$$

where $P_{load}$ is electricity demand, and $\eta_{inv}$ and $\eta_{Bat}$ are inverter efficiency and charge efficiency, respectively.

In order to meet load demand in case of a failure of RES, the energy stored in the batteries can be used. The amount of electricity discharged can be expressed by Equation (A9):

$$E_b(t) = E_b(t-1) - [P_{Load}(t)/\eta_{in\,v} - P_{Rs}(t)] \times \eta_{Bat} \tag{A9}$$

The given flowchart (Figure A2) depicts the situations that may occur during operation of the hybrid system. First, there is an excess of energy from RES and a need for storage, where the constraint of maximal capacity should be taken into account. Second, electricity produced from PV–WT–BB is equal to load demand, which is rarely the case. Last, RES fails to supply the load, and the developed algorithm considers the constraint of minimal depth of discharge and precisely calculates the amount of energy not covered (Eloss).

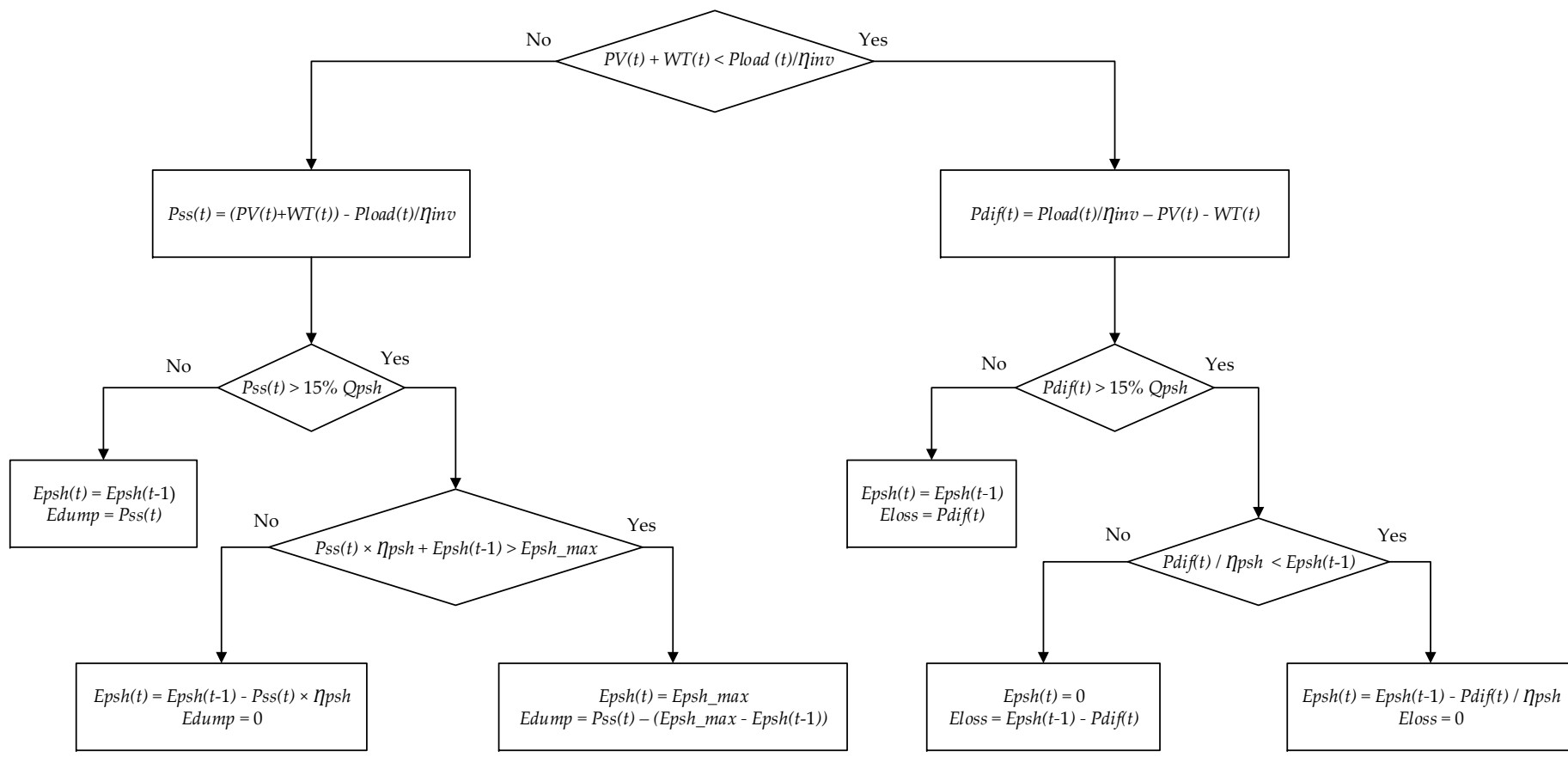

**Figure A1.** Energy flow of PV-WT-PSH.

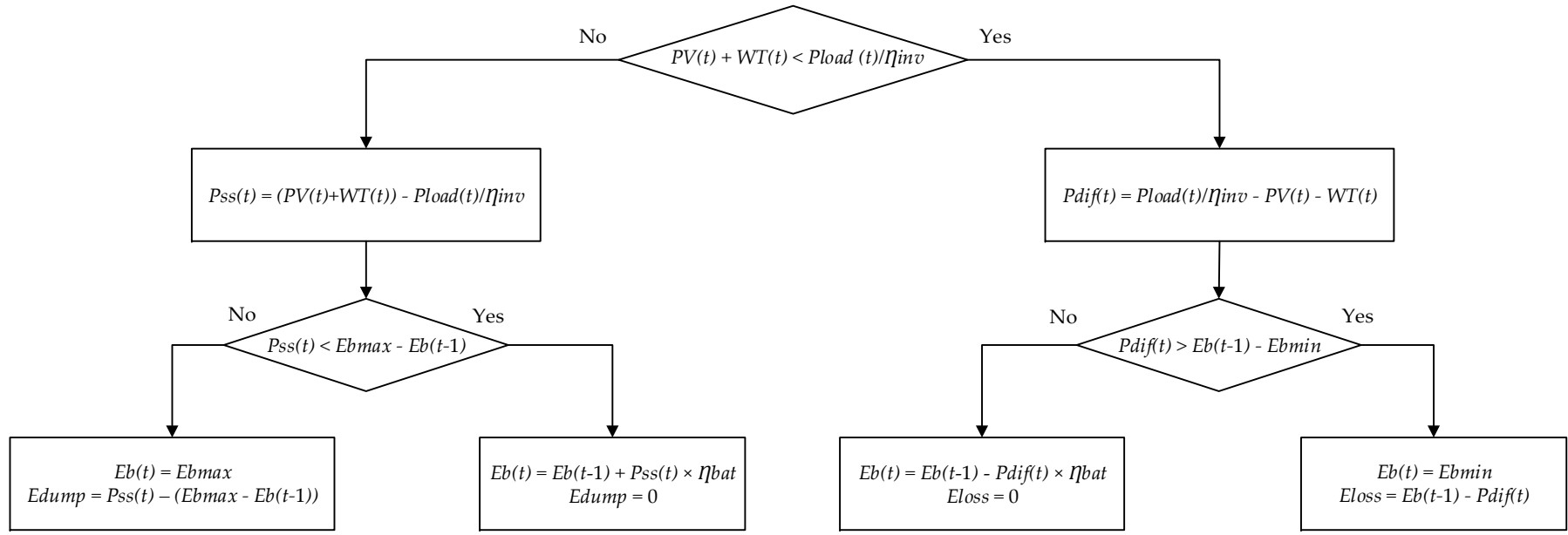

**Figure A2.** Energy management of PV-WT-BB.

## Appendix B. Net Present Cost Calculation

NPC$_{cp}$ can be calculated along these equations [39]:

$$NPC_{cp} = IC + RC + MC - C_{slv} \tag{A10}$$

where IC is the initial cost of the system and expressed as follows:

$$IC = P_{cp} \times C_{cp} \tag{A11}$$

where P$_{cp}$ [kW] is the optimal installed capacity. C$_{cp}$ is the capital cost (€/kW) that incorporates the owner's cost, civil, construction, mechanical equipment supply and installation expenses [41].

The replacement cost (RC) is evaluated based on Equation (A12):

$$RC = P_{cp} \sum_{j=1}^{N_{rep}} C_{cp}^R \times \frac{1}{(1+i)^{CLFT*j}} \tag{A12}$$

where $C_{cp}^R$ is the capital cost of a component at the time of replacement. N$_{rep}$—number of replacements over the lifetime of the project. CLFT—lifespan of the component.

Maintenance and operation cost (MC) is calculated based on the following formula:

$$MC = P_{cp} \sum_{z=1}^{PLFT} C_{cp}^{o\&m} \times \frac{1}{(1+i)^z} \tag{A13}$$

where $C_{cp}^{o\&m}$ (€/kW) is composed of fixed and variable costs related to maintaining and operating the component [40].

For the calculation of salvage value [39] the following Equations (A14)–(A16) apply:

$$C_{slv} = C_{cp}^R \times \frac{CR_{rem}}{CLFT} \left( \frac{1}{(1+i)^{PLFT}} \right) \tag{A14}$$

where CR$_{rem}$ is the remainder of the lifetime of the component at the end of the project lifespan:

$$CR_{rem} = CLFT - \left[ PLFT - LFTC_{rep} \right] \tag{A15}$$

$$LFTC_{rep} = CLFT \times \text{integer} \left[ \frac{PLFT}{CLFT} \right] \tag{A16}$$

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
