# Peer review of "Techno-Economic and Environmental Analysis of a Hybrid PV-WT-PSH/BB Standalone System Supplying Various Loads"

_energies, doi:10.3390/en12030514_

Round 1
Reviewer 1 Report
The subject paper presents a techno-economic and environmental analysis of solar-wind hybrid system with storage supplying in Mostaganem, Algeria. The specific comments are as follows.
1) Title is too long including a parenthesis. Please revise the title and also remove parenthesis.
2) How such a hybrid system would work? Is it an AC system or DC? How will load-generation balance be checked? Will it be based on frequency? Is the system connected to main grid? How will the hybrid system operate from power system point of view?
3) Fig.1 clearly indicates that there is no large-scale grid. Is the renewable energy target set for an isolated system?
4) What is the main contribution of this paper? Why such design as reported in the paper important. The authors fail to clarify this issue in the literature review and research gap.
5) Pumped storage hydro is an AC system. How is it being blend with PV and wind?
6) Why is Multi objective grey wolf optimizer chosen?
7) In 5.4: what is the reliability index being used?
8) The result section is too long. The reviewer does not see the need to show the performance of each load separately. Alternatively, the results could be presented in much concise form using table and figures.
9) The paper is too long without answering the real power system issues. The authors are requested to consider the above queries before further submission.
10) Where is the conclusion? Please provide a conclusion summarizing the main fainding of this paper.
Author Response
Dear Reviewer,
Please find our responses in the attached file.

Reviewer 2 Report
Abstract:
- rows 28-32 include this comment in the previous paragraph of the abstract. In general make the abstract more concise.
Introduction
- row 62 idiosyncratic: is it the right collocation of this word? a synonymous could be more effective for the context
- row 80 by (double), erase typo
Hybrid system and modelling
- row 149 definition of hybrid system: it is generally true when coupling two technologies with different features. If based on Renewables, this is additional.
- Equation 2: What do the constant values 20 and 800 stand for?
. Equaiton 6: Do you assume that there is no fluidodynamic interaction among the turbines?
Specify that each of them is independent.
- Modelling equations in section 2: you can be even more concise with equations in the manuscript and shift most of them to the appendix. Many equations show a high level of details. I would sugegst to keep methodological aspect in this part (that is optional, just a clue to improve the readability of the paper and the communciatione effectiveness).
Sizing bjective and optimization
- rows 218-219: LPSP definition is not clear: explain better
- row 221 above --> above-mentioned
- equation 17: P_load suggests the idea of power rather than energy.. maybe reconsider the notation.
- row 229: minus the salvage value ears --> what do you mean?
- Equation 21 and followings (with ref. to equation 18): for the purpose of this study interest and discount rate schould coincide. beside the formulation, Table 1 report rates with a different absolute value (8% and 3.5% respectively). Since this may markedly affect teh result of the case-study analysis, I ask to revise definition and all performed calculations.
- Table 1, battery costs: are costs of battery also dependant on the storage capacity?
- equations 27-28: put the right symbol (it is not Euros!)
- CO2 emissions (mass specific): measurment unit, more concise (CO2 as subscript)
Results
General comment about results presentation and discussion:
- improve the quality of the pictures
- think of aggregating some figures and to make a smart use of captions (there are many!)
Punctual comments:
- rows 305-306: rephrase, not clear
- sometimes the definition of storage "capacity" and the ability to supply "power" are misleading.
Moreover, power --> kW and (energy) capacity --> kWh. For example, at row 307, what are you indicating? Same observation regarding PHS so-called capacity at row 310
- row 324: edit "impacts patients' safety"
Discussion and conclusions
- Results summary at figure 23: You can highlight in the graph the surplus cost in each single user, with regard to the results obtained through the aggregation of loads.
- Figure 24: as in figure 23, plot also the co2 emissions related to the algerian power grid as a constant straight red line.
- row 596: edit "terms"
- row 601: edit "owing"
Author Response

(The authors gave the same response as above.)

Round 2
Reviewer 1 Report
No further comments.
Reviewer 2 Report
The authors carefully addressed the queries oof teh first review round.